METHODS AND RESOURCES

# A phylogenomic framework for charting the diversity and evolution of giant viruses

**Frank O. Aylward** [1,2]*, **Mohammad Moniruzzaman**[1], **Anh D. Ha**[1], **Eugene V. Koonin**[3]

**1** Department of Biological Sciences, Virginia Tech, Blacksburg, Virginia, United States of America, **2** Center for Emerging, Zoonotic, and Arthropod-borne Pathogens, Virginia Tech, Blacksburg, Virginia, United States of America, **3** National Center for Biotechnology Information, National Library of Medicine, National Institutes of Health, Bethesda, Maryland, United States of America

* faylward@vt.edu

**Data Availability Statement:** All data products described in this study are available on the Giant Virus Database: https://faylward.github.io/GVDB/. Reference trees of concatenated alignments can be

## Abstract

Large DNA viruses of the phylum Nucleocytoviricota have recently emerged as important members of ecosystems around the globe that challenge traditional views of viral complexity. Numerous members of this phylum that cannot be classified within established families have recently been reported, and there is presently a strong need for a robust phylogenomic and taxonomic framework for these viruses. Here, we report a comprehensive phylogenomic analysis of the Nucleocytoviricota, present a set of giant virus orthologous groups (GVOGs) together with a benchmarked reference phylogeny, and delineate a hierarchical taxonomy within this phylum. We show that the majority of Nucleocytoviricota diversity can be partitioned into 6 orders, 32 families, and 344 genera, substantially expanding the number of currently recognized taxonomic ranks for these viruses. We integrate our results within a taxonomy that has been adopted for all viruses to establish a unifying framework for the study of Nucleocytoviricota diversity, evolution, and environmental distribution.

## Main text

Large double-stranded DNA viruses of the phylum Nucleocytoviricota are a diverse group of viruses with virion sizes reaching up to 1.5 μm and genome sizes up to 2.5 Mb, comparable to many bacteria and archaea as well as picoeukaryotes [1–5]. The recognized taxonomic ranks in this phylum currently include 2 classes, 5 orders, 7 families, and 41 genera. The viruses in the families Asfarviridae, Ascoviridae, Iridoviridae, and Poxviridae infect metazoans, whereas those in the families Marseilleviridae, Mimiviridae, and Phycodnaviridae primarily infect algae or heterotrophic unicellular eukaryotes [6–8]. Members of the Nucleocytoviricota span an exceptionally broad range of genome sizes, from below 100 kbp to more than 2.5 Mbp. Several comparative genomic analyses have documented the highly complex, chimeric nature of their genomes in which numerous genes appear to have been acquired from diverse cellular lineages and other viruses [9–13]. These multiple, dynamic gene exchanges between viruses and their hosts [14–17] as well as the large phylogenetic breadth of this viral group [12,18,19] make the investigation of the evolution and taxonomic classification of the Nucleocytoviricota a challenging task. Despite these difficulties, early comparative genomic analyses studies succeeded in identifying a small set of core genes that could be reliably used to produce phylogenies that

found on the interactive Tree of Life: https://itol.
embl.de/shared/faylward.

**Funding:** This work was supported by a Simons
Early Career Award in Marine Microbial Ecology
and Evolution to F.O.A, an the NSF (IIBR-1918271)
award to F.O.A., and the Intramural Research
Program of the National Institutes of Health
(National Library of Medicine) for E.V.K. The
funders had no role in study design, data collection
and analysis, decision to publish, or preparation of
the manuscript.

**Competing interests:** The authors have declared
that no competing interests exist

**Abbreviations:** AaV, Aureococcus anophagefferens
virus; ANI, average nucleotide identity; ASFV,
African swine fever virus; A32, A32-like packaging
ATPase; GVOG, giant virus orthologous group;
HMM, Hidden Markov Model; IC, Internode
Certainty; ICTV, International Committee on the
Taxonomy of Viruses; MAG, metagenome-
assembled genome; MCP, major capsid protein;
OG, orthologous group; PolB, family B DNA
Polymerase; RED, relative evolutionary distance;
RNAPL, large RNA polymerase subunit; RNAPS,
small RNA polymerase subunit; SFII, superfamily II
helicase; TC, Tree Certainty; TetV, Tetraselmis
virus; TFIIB, TFIIB transcriptional factor; TopoII,
Topoisomerase family II; VLTF3, virus late
transcription factor 3.

encompass the entire diversity of Nucleocytoviricota, leading to the conclusion that all these
viruses share common evolutionary origins [18,20].

Recent studies have reported numerous new Nucleocytoviricota genomes, many of which
seem to represent novel lineages with only distant phylogenetic affinity for previously identi-
fied taxa [10,16,21]. For example, many viruses that infect a variety of protist genera have been
discovered that are related to Mimiviridae but do not fall within the same clade as the canoni-
cal *Acanthamoeba polyphaga mimivirus* [9,22,23]. Moreover, numerous metagenome-assem-
bled genomes (MAGs) have been reported that also appear to form novel sister clades to the
Mimiviridae, Asfarviridae, and other families [10,16,21]. Uncertainty in the phylogenetic rela-
tionships within the Nucleocytoviricota is a major impediment to the ongoing efforts that seek
to characterize the diversity of these viruses in the environment, as well as studies aiming to
better understand the evolutionary origins of unique traits within this viral phylum. As more
studies begin to chart the environmental diversity of Nucleocytoviricota, defining taxonomic
groupings that encompass equivalent phylogenetic breadths will be critical for the exploration
of the geographic and temporal variability in viral diversity and for comparing results from dif-
ferent studies. Moreover, the evolutionary origins of large genomes, virion sizes, and complex
metabolic repertoires in many Nucleocytoviricota are of great interest, and ancestral state
reconstructions and the tracking of horizontal gene transfers fully depend on a robust phyloge-
netic framework.

Here, we present a phylogenomic framework for charting the diversity and evolution of
Nucleocytoviricota. We first assess the strength of the phylogenetic signals from different
marker genes that are found in a broad array of distantly related viruses and arrive at a set of 7
genes that performs well in our benchmarking of concatenated protein alignments. Using this
hallmark gene set, we then perform a large-scale phylogenetic analysis and clade delineation of
the Nucleocytoviricota to produce a hierarchical taxonomy. Our taxonomy includes the estab-
lished families Poxviridae, Asfarviridae, Iridoviridae, Phycodnaviridae, Marseilleviridae, and
Mimiviridae as well as 26 proposed new family-level clades and 1 proposed new order. Sixteen
of the families are represented only by genomes derived from cultivation-independent
approaches, underscoring the enormous diversity of these viruses in the environment that
have not yet been isolated. We integrate these family-level classifications into the broader hier-
archical taxonomy of all viruses that has recently been adopted (i.e., a "megataxonomy" [3]) to
arrive at a unified and hierarchical classification scheme for the entire phylum
Nucleocytoviricota.

## Results

### Phylogenetic benchmarking of marker genes

We first generated a dataset of protein families to identify phylogenetic marker genes that are
broadly represented across Nucleocytoviricota. To this end, we selected a set of 1,380 quality-
checked Nucleocytoviricota genomes that encompassed all established families (S1 Data; see
Methods). By clustering the protein sequences encoded in these genomes, we then generated a
set of 8,863 protein families, which we refer to as giant virus orthologous groups (GVOGs).
We examined 25 GVOGs that were represented in >70% of all genomes and ultimately arrived
at a set of 9 GVOGs that were potentially useful for phylogenetic analysis, which is largely con-
sistent with the previous studies that have identified phylogenetic marker genes in Nucleocyto-
viricota [19,20,24] (Table 1, Figs A-Y in S1 Text, see Methods for details; descriptions of the 25
GVOGs provided in S2 Data). These GVOGs included 5 genes that we have previously used
for phylogenetic analysis of Nucleocytoviricota: the family B DNA Polymerase (PolB),
A32-like packaging ATPase (A32), virus late transcription factor 3 (VLTF3), superfamily II

**Table 1. Broadly represented GVOGs used for phylogenetic benchmarking.**

| GVOG ID | Name | Annotation |
| --- | --- | --- |
| GVOGm0003 | MCP | NCLDV major capsid protein |
| GVOGm0013 | SFII | DEAD/SNF2-like helicase |
| GVOGm0022 | RNAPS | DNA-directed RNA polymerase beta subunit |
| GVOGm0023 | RNAPL | DNA-directed RNA polymerase alpha subunit |
| GVOGm0054 | PolB | DNA polymerase family B |
| GVOGm0172 | TFIIB | Transcription initiation factor IIB |
| GVOGm0461 | TopoII | DNA topoisomerase II |
| GVOGm0760 | A32 | Packaging ATPase |
| GVOGm0890 | VLTF3 | Poxvirus Late Transcription Factor VLTF3 |

helicase (SFII), and major capsid protein (MCP) [10]. In addition, this set included the large and small RNA polymerase subunits (RNAPL and RNAPS, respectively), the TFIIB transcriptional factor (TFIIB), and the Topoisomerase family II (TopoII).

We evaluated individual marker genes and concatenated marker sets using the Internode Certainty and Tree Certainty metrics (IC and TC, respectively), which provide a measure of the phylogenetic strength of each individual marker gene [25,26]. The TC values were highest for the RNAP subunits, PolB, and TopoII (Fig 1A), consistent with the view that, in most cases, longer genes carry a stronger phylogenetic signal, likely due to the larger number of phylogenetically informative characters. A similar observation has also been made for phylogenetic marker genes of bacteria and archaea [27]. The MCP marker had markedly lower TC values than PolB, TopoII, or either of the RNAP subunits; this is potentially because Nucleocytoviricota genomes often encode multiple copies of MCP, which complicates efforts to distinguish orthologs from paralogs (Fig 1A). This is especially true when using metagenome-derived genomes that are incomplete, because orthologous MCP copies may be missing even while paralogs are present. When this occurs, a paralogous MCP will have the best match to this protein family and will be included even if it has experienced distinct evolutionary pressures compared to the orthologous copy. SFII, TFIIB, A32, and VLTF3 showed lower TC values than the other 5 markers, but these were also the shortest marker genes and would not be expected to yield high quality phylogenies when used individually.

Next, we sought to identify which marker genes provide for the best phylogenetic inference when used together in a concatenated alignment. If markers produce incongruent phylogenetic signals, they will yield trees with low TC values when concatenated, even if the individual phylogenetic strength of the markers is high [26]. We evaluated 8 marker gene sets in total. We began by assessing the TC of the 5-gene set that we have previously used [10]. Surprisingly, the TC of this set was lower than that of some individual markers (TC of 0.865; Fig 1B), suggesting that some of the markers provide incongruent signals. We surmised that this was most likely due to the MCP, given that the presence of multiple copies of this protein in some Nucleocytoviricota may complicate efforts to identify the appropriate ortholog to use for tree construction (Fig 1A). As we suspected, removal of MCP increased the TC of the concatenated tree (from 0.865 to 0.875) (Fig 1B). The addition of the RNAPS, RNAPL, TFIIB, or TopoII markers to the 4-gene set increased the TC (Fig 1B), although a 7-gene marker set that excluded RNAPS performed best overall (TC of 0.898). The existence of RNAPS paralogs has been observed before [23], and it is likely that this is the cause of the lower TC value when using this marker. Overall, the 7-gene marker set represents an improvement over the initial 5-gene set, and we therefore used these genes for subsequent phylogenetic analysis and clade demarcation. Importantly, the benchmarking results we present here are specific to the genome set that we analyzed, and the

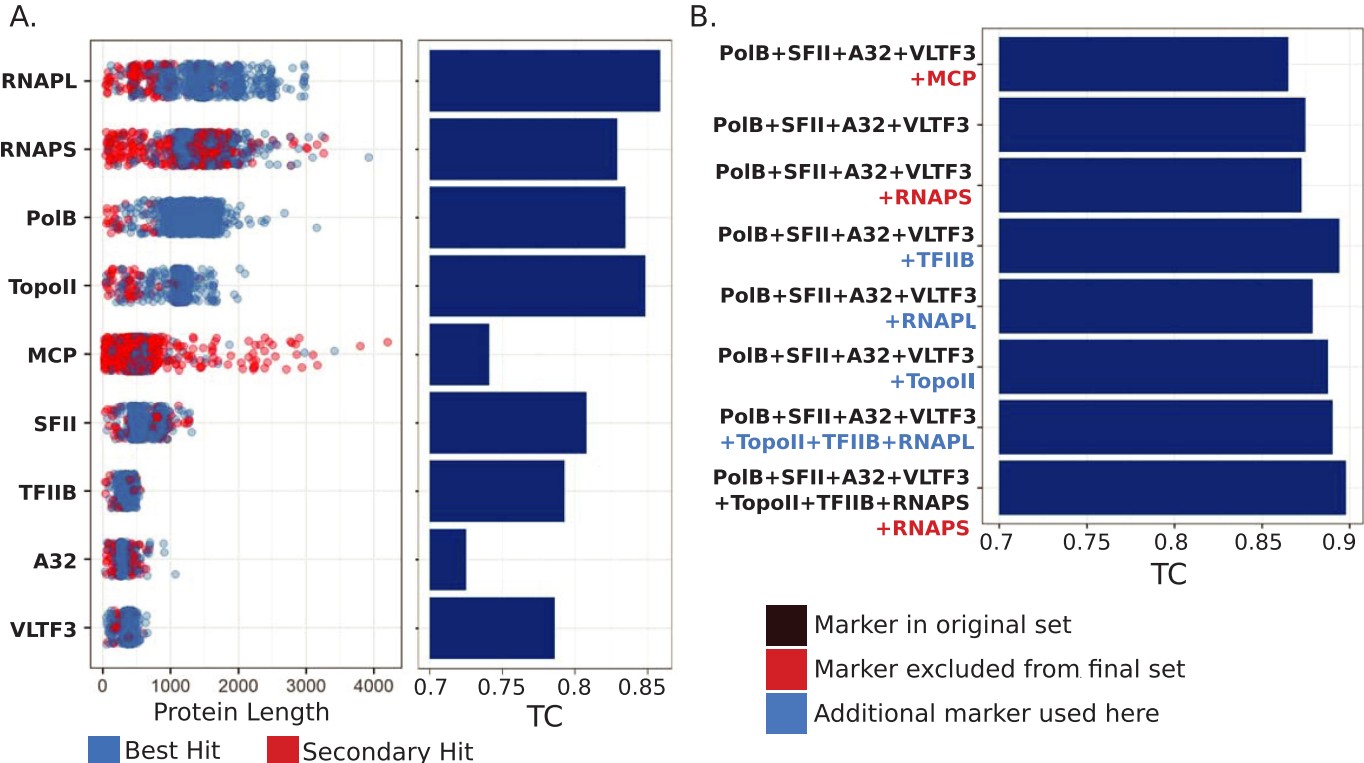

**Fig 1. Benchmarking of phylogenetic marker genes for Nucleocytoviricota.** (A) Dotplot of protein lengths for each of the 9 marker genes examined in detail. Blue dots represent proteins that were the best hit against marker gene HMMs and likely represent true orthologs, while red dots represent multiple copies of marker genes present in a genome. The TC scores of the markers are presented on the barplot on the right. (B) TC values for phylogenies made from concatenated alignments of different marker sets. Black text denotes markers we have used previously, red text denotes markers that we did not include in the final set, and blue text denotes additional markers used here compared to our original 5-gene set. Note that MCP was used in our original marker set but is excluded from the final 7-gene set. Protein lengths and TC values are provided in S2 Data. A32, A32-like packaging ATPase; HMM, Hidden Markov Model; MCP, major capsid protein; PolB, family B DNA Polymerase; RNAPL, large RNA polymerase subunit; RNAPS, small RNA polymerase subunit; SFII, superfamily II helicase; TC, Tree Certainty; TFIIB, TFIIB transcriptional factor; TopoII, Topoisomerase family II; VLTF3, virus late transcription factor 3.

use of MCP and RNAPS as phylogenetic markers may still be useful in other contexts. For example, when analyzing only complete genomes the presence of multiple paralogous copies of these genes may be less problematic.

## A hierarchical taxonomy for Nucleocytoviricota

The best-quality phylogenetic tree produced with the 7-gene marker set could be broadly divided into 2 class-level and 6 order-level clades, 5 of which were consistent with the orders in the recently adopted megataxonomy of viruses (Fig 2) [3]. The Chitovirales and Asfuvirales orders, which respectively contain the Poxviridae and Asfarviridae, formed a distinct group with a long stem branch (class Pokkesviricetes) that we used to root the tree, consistent with previous studies [20,28]. The Pimascovirales, which includes Pithoviruses, Marseilleviruses, and Iridoviridae/Ascoviridae, also formed a highly supported monophyletic group. The current order Algavirales, which includes the Phycodnaviridae, Chloroviruses, Pandoraviruses, Molliviruses, Prasinoviruses, and Coccolithoviruses, was paraphyletic, and we split this order into 2 groups based on their placement in the phylogeny. In the proposed taxonomy, we retain the existing Algavirales name for the clade that contains the Chloroviruses and Prasinoviruses and additionally propose the order pandoravirales for the group that includes the

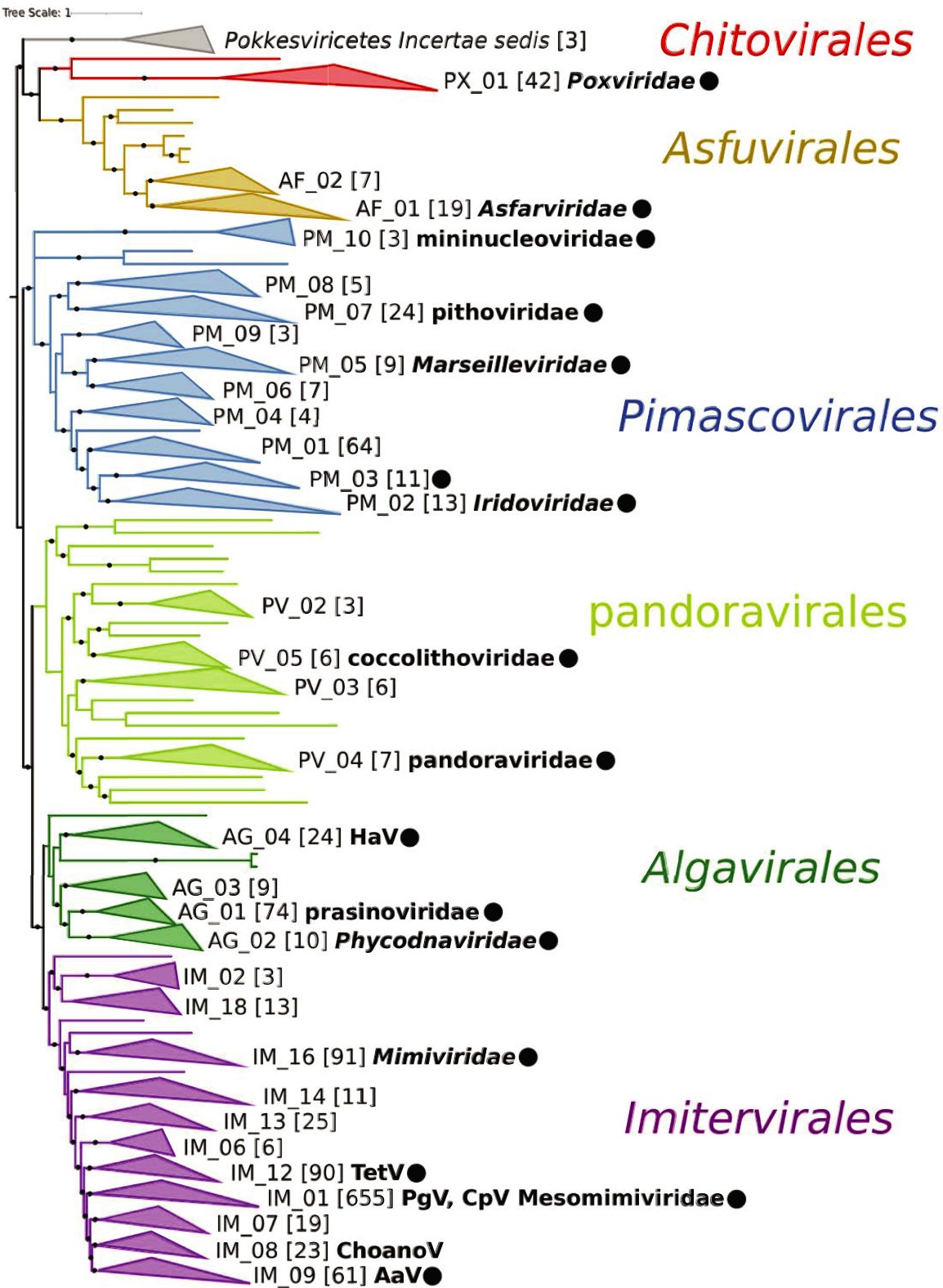

**Fig 2. Phylogeny of Nucleocytoviricota based on the 7-gene marker gene set that had the highest TC value of those tested.** The phylogeny was inferred using the LG+I+F+G4 model in IQ-TREE. Solid circles denote IC values >0.5. Families are denoted by collapsed clades, with their nonredundant identifier provided at their right. The number of genomes in each clade is provided in brackets. Established family names are provided in bold italics, and proposed names are provided in lowercase. The presence of notable cultivated viruses is provided in bold next to some clades. Aav, *Aureococcus anophagefferens* virus; ChoanoV1, Choanoflagellate virus; CpV, *Chrysochromulina parva* virus; HaV, *Heterosigma akashiwo* virus; IC, Internode Certainty; PgV, *Phaeocystis globosa* virus; TC, Tree Certainty; TetV, *Tetraselmis* virus.

Pandoraviruses and Coccolithoviruses. The Imitervirales, which contain the Mimiviridae, formed a sister group to the Algavirales.

From our reference tree, we delineated taxonomic levels using the relative evolutionary distance (RED) of each clade as a guide, using an approach similar to the one recently employed for bacteria and archaea [29]. RED values vary between 0 and 1, with lower values denoting phylogenetically broad groups that branch closer to the root and higher values denoting phylogenetically shallow groups that branch closer to the leaves. The RED of the Nucleocytoviricota classes ranges from 0.017 to 0.032, whereas the values for the orders range from 0.158 to 0.240 (Fig 3A and S3 Data). We delineated family- and genus-level clades so that they had nonoverlapping RED values that were higher than their next-highest taxonomic rank (Fig 3A). This approach yielded clades that were consistent with families and genera currently recognized by the International Committee on Taxonomy of Viruses (ICTV; [30]), such as the *Chlorovirus*, *Prasinovirus*, and Mimiviridae (see below; full classification information in S1 Data). To ensure that putative families were not defined by spurious placement of individual genomes, we accepted only groups with ≥3 members and left other genomes in the tree as singletons with incertae sedis as the family identifier. This approach yielded a total of 32 families, not including 22 singleton genomes that potentially represent additional families and are listed as incertae sedis here. We provided tentative genus-level identifiers for all genomes, leading to 344 total genera (Figs 2 and 3). Of these, 213 genera contain only a single representative, and additional merging or splitting of these groups may be necessary as more genomes become available and fine-scale phylogenetic patterns are clarified.

Of the 32 families, 6 correspond to the families currently recognized by the ICTV, for which we retained the existing nomenclature (Asfarviridae, Poxviridae, Marseilleviridae, Iridoviridae, Phycodnaviridae, and Mimiviridae). The Ascoviridae are included within the Iridoviridae, and so we use the latter family name here. In addition, we propose 6 family names here: "prasinoviridae," which include the prasinoviruses, "pandoraviridae," which include the Pandoraviruses and *Mollivirus sibericum*, "coccolithoviridae," which include the coccolithoviruses, "pithoviridae," which include Pithoviruses, Cedratviruses, and Orpheoviruses, "mesomimiviridae," which includes several haptophyte viruses previously defined as "extended Mimiviridae," and "mininucleoviridae," which has previously been described and includes several viruses of Crustacea [31]. Some of these family names have been used previously, such as pandoraviridae and mininucleoviridae, but so far have not been formally recognized by the ICTV. For other proposed families, we provide nonredundant identifiers corresponding to their order, and we anticipate that future studies will provide information for selecting appropriate family names once more is learned on the host ranges and molecular traits of these viruses. Two of the families contained only a single cultivated representative (AG_04 and IM_09), whereas 16 families included none.

Notably, the Imitervirales contain 11 families, as well as 4 singleton viruses that potentially represent additional family-level clades. This underscores the vast diversity of the large viruses in this group, which is consistent with the results of several studies reporting an enormous diversity of Mimiviridae-like viruses in the biosphere, in particular in aquatic environments [10,32–34]. Other studies have suggested additional nomenclature to refer to these

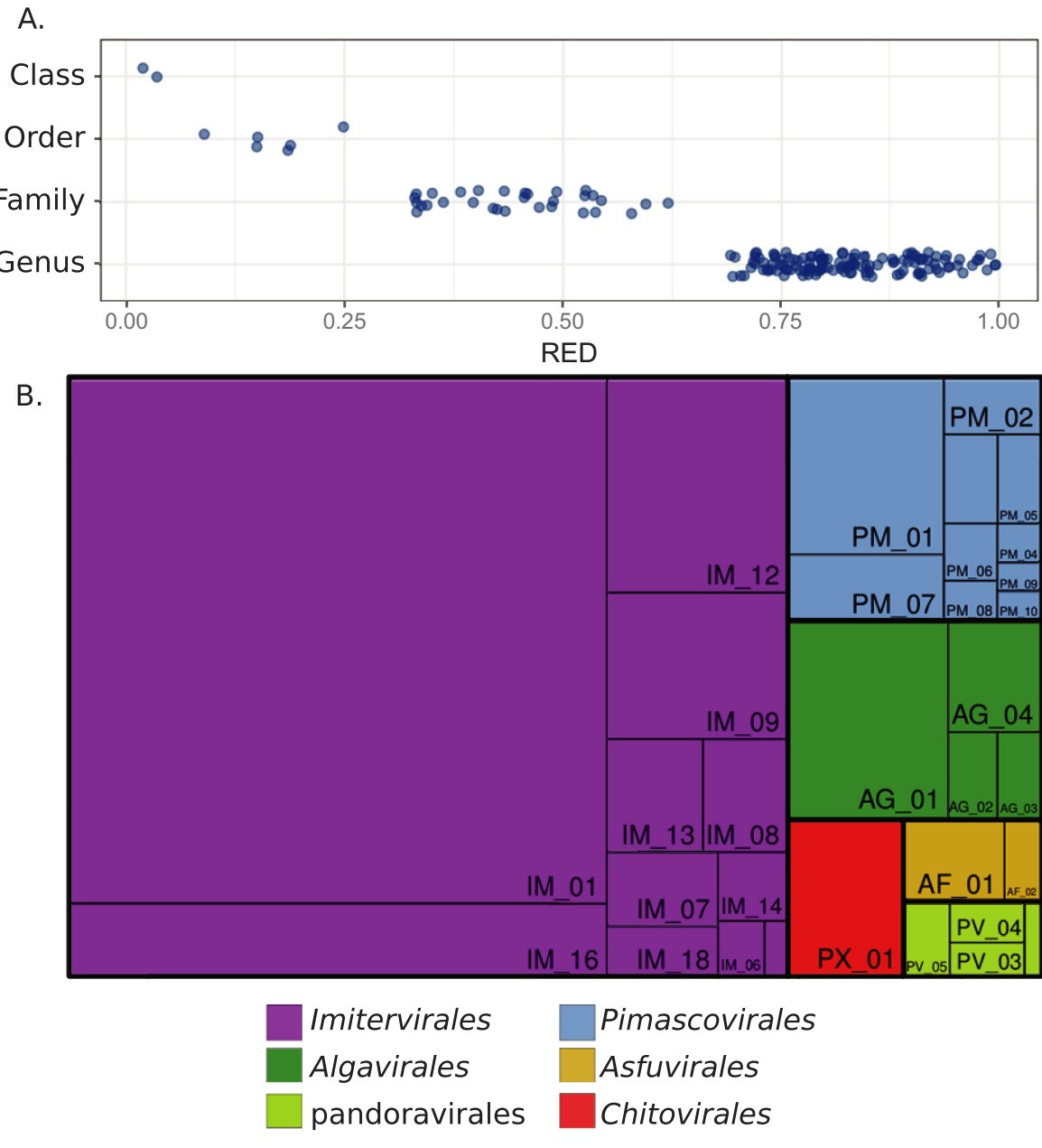

**Fig 3. Summary of the Nucleocytoviricota taxonomy.** (A) RED values for Nucleocytoviricota classes, orders, and families, and genera. (B) Treemap diagram of the Nucleocytoviricota in which orders and families are shown. The area of each rectangle is proportional to the number of genomes in the respective taxon. RED values can be found in S3 Data. RED, relative evolutionary divergence.

Mimiviridae-like viruses, such as the "extended Mimiviridae" and the subfamilies Mesomimivirinae, or Megamimivirinae, but our results suggest that an extensive array of new families is warranted within Imitervirales, given the broad genomic and phylogenetic diversity within this group. Several of the proposed new families contain representatives that have recently been described; IM_12 contains the *Tetraselmis* virus (TetV), which encodes several fermentation genes [11], IM_09 contains *Aureococcus anophagefferens* virus (AaV), which is thought to play an important role in brown tide termination [35], and IM_08 contains a virus of

Choanoflagellates [36] (Fig 2). Family IM_01 contains cultivated viruses that infect hapto-phytes of the genera *Chrysochromulina* and *Phaeocystis*, which were previously proposed to be classified in the subfamily mesomimivirinae [23]. We propose the name mesomimiviridae to denote the family-level status of this lineage, while still retaining reference to this original name. Notably, the Mesomimiviridae includes by far the largest total number of genomic representatives in our analysis (*n* = 655, including 652 MAGs; Figs 2 and 3B), the vast majority of which are derived from aquatic environments (Fig Z in S1 Text), suggesting that members of this family are important components of global freshwater and marine ecosystems. Within the Mimiviridae, we recovered 3 clades that correspond to previously proposed subfamilies. One of these clades contains Klosneuviruses and corresponds to the proposed subfamily Klosneu-virinae [37]; this subfamily also includes *Bodo saltans* virus as well as several genomes recovered from forest soils [38,39]. The second clade corresponds to the subfamily Megamimivirinae and includes *A. polyphaga mimivirus*, Tupanviruses, and *Megavirus chilensis*, among others [40–42]. Lastly, we recovered a clade that includes *Cafeteria roenbergensis* virus [9], several "PacV" viruses obtained from flow sorting and sequencing of marine samples [43], and a variety of MAGs.

All families within the Imitervirales except one included members with genome sizes >500 kbp, highlighting the "giant" genomes that are characteristic of this lineage (Fig 4A). Genes involved in translation, including tRNA synthetases and translation initiation factors, were consistently highly represented in the Imitervirales, showing that the rich complement of these genes that has been described for the Mimiviridae is broadly characteristic of other families in this order (Fig 4B) [40,42]. Throughout the Imitervirales genes involved in glycolysis and the TCA cycle, cytoskeleton components such as viral-encoded actin, myosin, and kinesin proteins, and nutrient transporters including those that target ammonia and phosphate were also common (Fig 4B) [10,44–46], underscoring the complex functional repertoires of this virus order.

The Algavirales is a sister lineage to the Imitervirales that contains 4 families encompassing several well-studied algal viruses. The Prasinoviridae (AG_01) is a family that includes viruses known to infect the prasinophyte genera *Bathycoccus*, *Micromonas*, and *Ostreococcus* [8], and cultivation-independent surveys have provided evidence that the MAGs in this clade are also associated with prasinophytes [46]. Similarly, our approach yielded a well-defined Phycodna-viridae family (AG_02) composed mostly of chloroviruses, consistent with the similar host range of these viruses [47]. All 4 families of the Algavirales have smaller genome sizes compared to the Imitervirales (Fig 4A), but there were still several similarities in their encoded functional repertoires. As noted previously [10,17,36], genes involved in light sensing, including rhodopsins and chlorophyll-binding proteins, were common across the Imitervirales and Algavirales, perhaps because many of the viruses are found in sunlit aquatic environments where manipulation of host light sensing during infection is advantageous. Moreover, genes involved in nutrient transport, translation, and even some components of glycolysis and the TCA cycle were found in the Algavirales, consistent with the complex repertoires of metabolic genes that have been reported for some of these viruses despite their relatively small genome sizes [48,49].

The pandoravirales, a new order we propose here, consists of 4 families, including the pandoraviridae and the coccolithoviridae. The pandoraviridae (PV_04) include *Mollivirus sibericum* as well as the pandoraviruses, which possess the largest viral genomes known [50]. Grouping of these viruses together in the same family is consistent with previous studies that have shown that *M. sibericum* and the Pandoraviruses have shared ancestry [51,52], and comparative genomic analysis that have shown that they all encode a unique duplication in the gly-cosyl hydrolase that has been co-opted as a major virion protein in the Pandoraviruses [53].

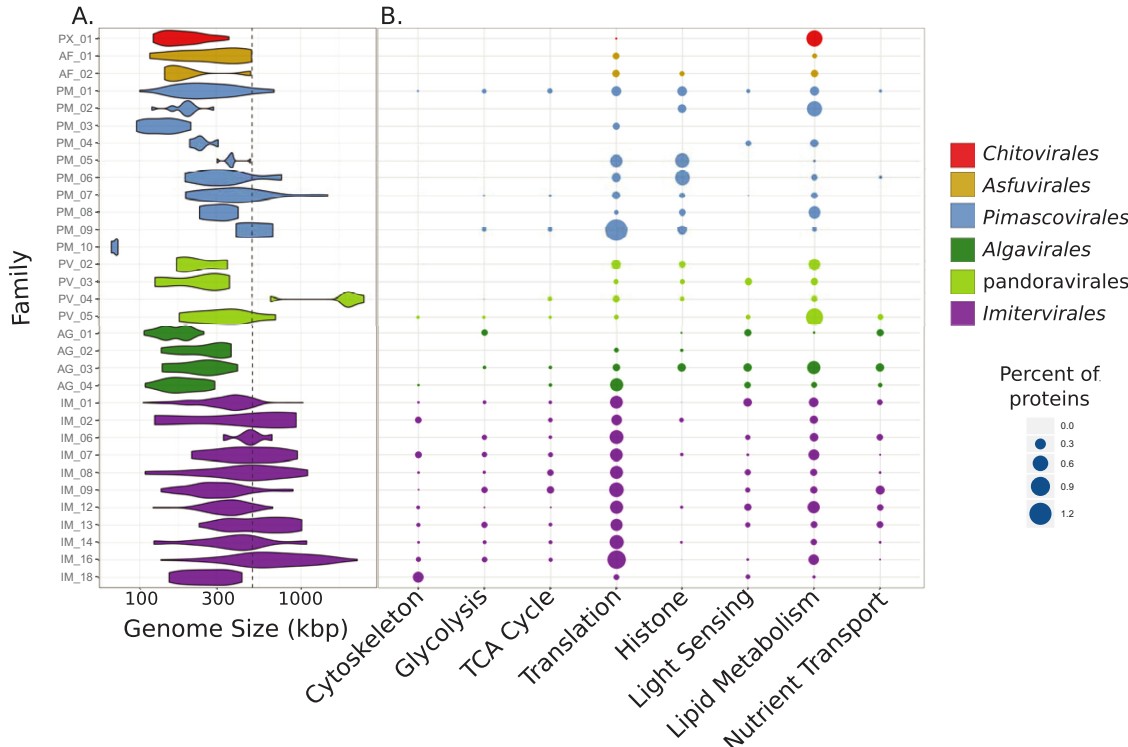

**Fig 4. Genomic characteristics of the Nucleocytoviricota.** (A) Violin plot showing the genome size distribution across the Nucleocytoviricota families. The dashed gray line denotes 500 kbp. (B) Bubble plot showing the percent of total proteins in each family that could be assigned to GVOGs that belonged to particular functional categories (details in S2 Data). GVOG, giant virus orthologous group.

The coccolithoviridae (PV_05) is mostly comprised of viruses that infect the marine cocco-lithophore *Emiliania huxleyi*; although much smaller than the genomes of the Pandoraviruses, genomes of cultivated representatives of this family exceed 400 kbp and encode diverse functional repertoires including sphingolipid biosynthesis genes [54].

Although most orders contained primarily genomes that could be readily grouped into families, the pandoravirales also included 15 singleton genomes out of the 37 total. This is potentially due to the lack of adequate genome sampling in this group, which would result in many distinct lineages represented by only individual genomes. If this is the case, more well-defined families will become evident as additional genomes are sequenced. Alternatively, the lack of clearly defined families could result from longer branches in this group that obfuscate the clustering of well-defined groups. The Medusavirus, which is included in this order, encodes a divergent PolB marker gene that is likely the result of gene transfer with a eukaryotic homolog [55]. Frequent gene transfers among phylogenetic marker genes might be another explanation for the presence of many long branches in the pandoravirales clade.

The Pimascovirales encompass 10 families including the Iridoviridae (PM_02), Marseille-viridae (PM_05), and Pithoviridae (PM_07) and notably includes both *Pithovirus sibericum*, which has the largest viral capsid currently known (1.5 μm [56]), as well as crustacean viruses in the family Mininucleoviridae (PM_10), which possess the smallest genomes recorded for any Nucleocytoviricota (67 to 71 kbp [31]). The Mininucleoviridae have highly degraded genomes that lack several phylogenetic marker genes. Although they can be classified within

the Pimascovirales with high confidence, their relationship to other families is uncertain, and we therefore placed them in a polytomous node at the base of this order (Fig 2). The uncharacterized family PM_01 contains the largest number of genomes (*n* = 64) within this order, all of which are MAGs. The majority of these MAGs were derived from aquatic metagenomes, and some have been recovered in marine metatranscriptomes [46], suggesting that they play an important but currently unknown role in marine systems. Overall, the repertoires of encoded proteins in the Pimascovirales were notably different from the Imitervirales, pandoravirales, and Algavirales; while cytoskeleton components, nutrient transporters, light sensing genes, and central carbon metabolism components were prevalent in the latter 3 families, they were largely absent in the Pimascovirales (Fig 4B). Conversely, histone components appeared to be more prevalent in the latter order; indeed, the histones encoded in marseilleviruses have recently become a model for understanding their structure and interactions with viral DNA [57,58]. Genes involved in translation and lipid metabolism were present in the Pimascovirales in addition to most other orders.

In addition to the families that fall within the established orders and families, we also identified several lineages or individual genomes that may represent novel taxonomic ranks (Fig 2). One of these groups consists of 3 genomes that is basal-branching to the Pokkesviricetes class, which we refer to as Pokkesviricetes incertae sedis (Fig 2). The basal-branching placement of this group suggests that it might comprise a new class that is a sister group to the Pokkesviricetes. The placement of this lineage remains tentative, however, and to clarify evolutionary relationships within the Nucleocytoviricota further phylogenetic work with additional genomes will be necessary both for this lineage as well as other putative novel taxa that are represented by individual genomes.

## Discussion

Although only 6 families of Nucleocytoviricota have been established to date, recent cultivation-independent studies have revealed a vast diversity of these viruses in the environment, and their classification, together with cultivated representatives, has remained challenging. Here, we present a unified taxonomic framework based on a benchmarked set of phylogenetic marker genes that establishes a hierarchical taxonomy of Nucleocytoviricota. This taxonomy encompasses 6 orders and 32 families, including 1 order and 26 families we propose here. Remarkably, the Imitervirales contain 11 families, including the Mimiviridae, underscoring the vast diversity of large viruses within this order. This framework substantially increases the total number of Nucleocytoviricota families, and we expect that the number will continue to increase as new genomes are incorporated. In particular, we identified 22 singleton genomes that likely represent additional families, the status of which will be clarified as more genomes become available.

We anticipate that the phylogenetic and taxonomic framework we develop here will be a useful community resource for several future lines of inquiry into the biology of Nucleocytoviricota. Firstly, the GVOGs are a large set of viral protein families constructed using many recently produced Nucleocytoviricota MAGs, and they will likely be useful for the genome annotation and the examination of trends in gene content across viral groups. Secondly, the reference phylogeny we present will facilitate work that delves into ancestral Nucleocytoviricota lineages, examines the timing and nature of gene acquisitions, and classifies newly discovered viruses. For example, giant viral genomes (>500 kbp) evolved independently in multiple orders, and future studies that examine the similarities and differences in these genome expansion events will be important for pinpointing the driving forces of viral gigantism. Lastly, analysis of the environmental distribution of different taxonomic ranks of Nucleocytoviricota

across Earth's biomes will be an important direction for future work that reveals prominent biogeographic patterns and helps to clarify the ecological impact of these viruses.

## Methods

### Nucleocytoviricota genome set

We compiled a set of Nucleocytoviricota genomes that included MAGs as well as genomes of cultured isolates. For this, we first downloaded all MAGs available from several recent studies [10,16,21]. We also included all Nucleocytoviricota genomes available in NCBI RefSeq as of June 1, 2020. Lastly, we also included several Nucleocytoviricota genomes from select publications that were not yet available in NCBI, such as the cPacV, ChoanoV, *Pyramimonas orientalis* virus O1B (MT663543), and AbALV viruses that have recently been described [15,36,43,59]. After compiling this set, we dereplicated the genomes, since the presence of highly similar or identical genomes is not necessary for broad-scale phylogenetic inference. For dereplication, we compared all genomes against each other using MASH v. 2.0 [60] ("mash dist" parameters -k 16 and -s 300), and clustered genomes together using a single-linkage clustering, with all genomes with a MASH distance of ≤0.05 linked together. The MASH distance of 0.05 was chosen since it has been roughly found to correspond to an average nucleotide identity (ANI) of 95% [60]; although gene flow can occur over a broad range of genome identity values [61], this is still a useful threshold for genome dereplication. From each cluster, we chose the genome with the highest N50 contig length as the representative. We then decontaminated the genomes through analysis with ViralRecall v.2.0 [62] (-c parameter), with all contigs with negative scores removed on the grounds that they represent non-Nucleocytoviricota contamination or highly unusual gene composition that cannot be validated by our present knowledge of Nucleocytoviricota genomic content. We only considered contigs >10 kbp, given the inherent difficulty in eliminating contamination derived from short contigs. To ensure that we only used genomes that could be placed in a phylogeny, we then screened the genome set and retained only those with a PolB marker and 3 of the 4 markers A32, SFII, VLTF3, and MCP, consistent with our previous methodology [10]. After this, we arrived at a set of 1,380 genomes, including 1,253 MAGs and 127 complete genomes of cultivated viruses.

### GVOG construction

To construct GVOGs, we first predicted proteins from all genomes using Prodigal v. 2.6.2. Proteins that did not have a recognizable start or stop codon at the ends of contigs were removed on the grounds that they may represent fragmented genes and obfuscate orthologous group (OG) predictions. We then calculated OGs using Proteinortho v. 6.06 [63] (parameters -e = 1e-5—identity = 25 -p = blastp+—selfblast—cov = 50 -sim = 0.80). We constructed Hidden Markov Models (HMMs) from proteins by aligning them with Clustal Omega v1.2.3 [64] (default parameters), trimming the alignment with trimAl v1.4.rev15 [65] (parameters -gt 0.1), and generating the HMM from the trimmed alignment with hmmbuild in HMMER v3.3 [66]. The goal of this analysis was to identify broad-level protein families, and we therefore sought to merge HMMs that bore similarity to each other and therefore derived from related protein families. For this, we then compared the proteins in each OG to the HMM of every other OG (hmmsearch -E 1e-20—domtblout option, hits retained only if 30% of the query protein aligned to the HMM). In cases where >50% of the proteins in one OG also had hits to the HMM of another OG, and vice versa, we then merged all of the proteins together and constructed a new merged HMM from the full set of proteins. The final set contained 8,863 HMMs, and we refer to these as the GVOGs. To provide annotations for GVOGs, we compared all of the proteins in each GVOG to the EggNOG 5.0 [67], Pfam [68], and NCVOG

databases [69] (hmmsearch, -E 1e-3). For NCVOGs, we obtained protein sequences from the original NCVOG study and generated HMMs using the same methods we used for GVOGs. Annotations were assigned to a GVOG if >50% of the proteins used to make a GVOG had hits to the same HMM in one of these databases. Details regarding all GVOGs and their annotations can be found in S2 Data.

## Benchmarking phylogenetic marker genes for Nucleocytoviricota

To identify phylogenetic markers for Nucleocytoviricota, we cataloged GVOGs that were broadly represented in the 1,380 viral genomes that we used for benchmarking. We searched all proteins encoded in the genomes against the GVOG HMMs using hmmsearch (e-value cutoff 1e-10) and identified a set of 25 GVOGs that were found in >70% of the genomes in our set (hmmsearch, -E 1e-5). We constructed individual phylogenetic trees of these protein families to assess their individual evolutionary histories. For individual phylogenetic trees, we calibrated bit score cutoffs so that poorly matching proteins would not be included. These cutoffs were generally equivalent to the fifth percentile score of all of the best protein matches for each genome. We then examined several features of these trees. Firstly, we only considered GVOGs present in all established families that would therefore be useful as universal or nearly universal phylogenetic markers. Secondly, we examined each tree individually to assess the degree to which taxa from different orders clustered together in distinct monophyletic groups, which was taken as a signature of HGT. High levels of gene transfer would produce topologies incongruent with other marker genes and therefore compromise the reliability of a given marker when used on a concatenated alignment. For individual marker gene trees, we aligned proteins from each GVOG using Clustal Omega, trimmed the alignment using trimAl (-gt 0.1 option), and constructed the phylogeny using IQ-TREE with ultrafast bootstraps calculated (-m TEST, -bb 1000, -wbt options).

We arrived at a set of 9 GVOGs that met the criteria described above and could potentially serve as robust phylogenetic markers (Table 1). We evaluated the phylogenetic strength of these markers individually using the recently developed TC and IC metrics. These metrics are an alternative to the traditional bootstrap because they take into account the frequency of contrasting bipartitions and can therefore be viewed as a measure of the phylogenetic strength of a gene [25,26]. We generated alignments using Clustal Omega, trimmed with TrimAl, and generated trees with IQ-TREE v1.6.9 [70] with ultrafast bootstraps [71] (parameters -wbt -bb 1000 -m LG+I+G4). We calculated TC and IC values in RaxML v8.2.12 (-f i option, ultrafast bootstraps used with the -z flag) [72]. We also evaluated the TC and IC values of trees generated from concatenated alignments. To construct concatenated alignments, we used the python program "ncldv_markersearch.py" that we developed for this purpose: https://github.com/faylward/ncldv_markersearch.

For the final tree used for clade demarcation, we ran IQ-TREE 5 times using the parameters "-m LG+F+I+G4 -bb 1000 -wbt," and we chose the resulting tree with the highest TC value for subsequent clade demarcation and RED calculation. Three genomes in the Mininucleoviridae family were included in the final tree but were not used for the benchmarking analysis because they have been shown to have highly degraded genomes that are not necessarily representative of Nucleocytoviricota more broadly [31]. Moreover, the MAG ERX555967.47 was found to have highly variable placement in different orders in different trees we analyzed, and we therefore did not include this genome in the final tree on the grounds that it represented a rogue taxa that may reduce overall tree quality [73]. We rooted the final tree between the Pokkesviricetes and Megaviricetes, consistent with previous studies [6,28]. We placed the 3 genomes of Pokkesviricetes incertae sedis adjacent to the Pokkesvirictes clade due to the clustering of

several GVOGs of this group with members of the Pokkesvirictes (SFII: Fig C in S1 Text, PolB: Fig I in S1 Text).

## Family delineation and nomenclature

We calculated RED values in R using the get_reds function in the package "castor" [74]. As input, we used a rooted tree derived from the 7-gene marker set described above. For the Poxviridae, Asfarviridae, Iridoviridae, Phycodnaviridae, Marseilleviridae, mininucleoviridae, and Mimiviridae, we retained existing nomenclature, and clades assigned these names based on the initially characterized viruses that were assigned to these families. For example, the Phycodnaviridae was assigned to AG_02 because the chloroviruses within this clade were the first-described members of this family, while the prasinoviruses were assigned to a new family, although they are commonly referred to as Phycodnaviridae. Similarly, Mimiviridae was assigned based on the placement of *A. polyphaga mimivirus*, Iridoviridae was assigned based on the placement of *Invertebrate iridescent virus 6*, Asfarviridae was assigned to the clade containing African swine fever virus (ASFV), and Marseilleviridae was assigned to the clade containing the marseilleviruses. The treemap visualization was generated using the R package "treemap."

## Supporting information

**S1 Text. Supporting figures.** Fig A. Major Capsid Protein GVOGm0003 phylogeny. Fig B. Disulfide (thiol) oxidoreductase GVOGm0004 phylogeny. Fig C. Superfamily II helicase GVOGm0013 phylogeny. Fig D. Patatin phospholipase GVOGm0018 phylogeny. Fig E. DEAD/SNF2-like helicase GVOGm0020 phylogeny. Fig F. DNA-directed RNA polymerase subunit beta (RNAPS) GVOGm0022 phylogeny. Fig G. DNA-directed RNA polymerase subunit alpha (RNAPL) GVOGm0023 phylogeny. Fig H. mRNA capping enzyme GVOGm0036 phylogeny. Fig I. DNA polymerase family B GVOGm0054 phylogeny. Fig J. TATA box binding protein (TBP) GVOGm0056 phylogeny. Fig K. Ribonucleoside diphosphate reductase, alpha subunit GVOGm0088 phylogeny. Fig L. D5-like helicase-primase GVOGm0095 phylogeny. Fig M. Uncharacterized, C-terminal domain GVOGm0115 phylogeny. Fig N. Uncharacterized protein GVOGm0152 phylogeny. Fig O. Transcription initiation factor IIB GVOGm0172 phylogeny. Fig P. RuvC, Holliday junction resolvases (HJRs) GVOGm0189 phylogeny. Fig Q. Ubiquitin carboxyl-terminal hydrolase GVOGm0214 phylogeny. Fig R. Proliferating cell nuclear antigen GVOGm0239 phylogeny. Fig S. DNA topoisomerase II GVOGm0461 phylogeny. Fig T. Divergent DNA-directed RNA polymerase subunit 5 GVOGm0694 phylogeny. Fig U. Packaging ATPase GVOGm0760 phylogeny. Fig V. Metallopeptidase WLM GVOGm0787 phylogeny. Fig W. Ribonuclease III GVOGm0798 phylogeny. Fig X. Virus Late Transcription Factor 3 VLTF3 GVOGm0890 phylogeny. Fig Y. Ribonucleotide reductase small subunit GVOGm1574 phylogeny. Fig Z. Barchart of source habitats for the Nucleocytoviricota families. Full information is provided in S1 Data. (PDF)

**S1 Data. Taxonomy, genome statistics, and other metadata for the Nucleocytoviricota genomes analyzed in this study.** (XLSX)

**S2 Data. Statistics and descriptions of the 25 GVOGs present in 70% of the genomes analyzed.** Full annotations of all GVOGs are also provided, and TC values for the trees of this study. (XLSX)

**S3 Data. RED values for taxonomic ranks presented in this study.**
(XLSX)

## Acknowledgments

We acknowledge the use of the Virginia Tech Advanced Research Computing Center for bio-informatic analyses performed in this study.

## Author Contributions

**Conceptualization:** Frank O. Aylward.

**Data curation:** Frank O. Aylward, Anh D. Ha.

**Formal analysis:** Frank O. Aylward, Mohammad Moniruzzaman, Anh D. Ha.

**Funding acquisition:** Frank O. Aylward.

**Investigation:** Frank O. Aylward, Mohammad Moniruzzaman.

**Methodology:** Frank O. Aylward, Mohammad Moniruzzaman, Anh D. Ha.

**Project administration:** Frank O. Aylward.

**Resources:** Frank O. Aylward, Eugene V. Koonin.

**Software:** Frank O. Aylward.

**Supervision:** Frank O. Aylward, Eugene V. Koonin.

**Validation:** Frank O. Aylward.

**Visualization:** Frank O. Aylward.

**Writing – original draft:** Frank O. Aylward, Eugene V. Koonin.

**Writing – review & editing:** Frank O. Aylward, Eugene V. Koonin.

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
