## [Editor Report · Decision Letter 0]

24 Jun 2021

Dear Dr Aylward, 

Thank you for submitting your manuscript entitled "A Phylogenomic Framework for Charting the Diversity and Evolution of Giant Viruses" for consideration as a Methods and Resources by PLOS Biology.

Your manuscript has now been evaluated by the PLOS Biology editorial staff, as well as by an academic editor with relevant expertise, and I'm writing to let you know that we would like to send your submission out for external peer review. Please accept my apologies for the delay incurred while we sought external advice.

Please re-submit your manuscript within two working days, i.e. by Jun 28 2021 11:59PM.

Kind regards,

Roli Roberts

Roland G Roberts PhD

Senior Editor

PLOS Biology

rroberts@plos.org

on behalf of

Paula Jauregui, PhD

Editor

PLOS Biology

---

## [Decision Letter · Decision Letter 1]

13 Aug 2021

Dear Dr. Aylward,

Thank you very much for submitting your manuscript "A Phylogenomic Framework for Charting the Diversity and Evolution of Giant Viruses" for consideration as a Methods and Resources at PLOS Biology. Your manuscript has been evaluated by the PLOS Biology editors, an Academic Editor with relevant expertise, and by several independent reviewers.

In light of the reviews (below), we are pleased to offer you the opportunity to address the comments from the reviewers in a revised version that we anticipate should not take you very long. We will then assess your revised manuscript and your response to the reviewers' comments and we may consult the reviewers again. Please also address the formatting and editorial policy requirements.

In particular, reviewer #1 wants you to provide the number of currently recognized viral genera in the phylum Nucleocytoviricota, and to explain “megataxonomy”, disagrees with the attribution the low TC value for the MCP tree to the existence of paralogs, says that the choice of ortholog is needed, has questions about the root of the phylogenetic tree, suggests you to describe your standpoint about Klosneuvirinae, wants you to add a few words about the enrichment of Translation genes in the PM_09, thinks that you should distinguish the transfer from medusavirus and eukaryotes, wants you to explain the classification of Pimascovirales and Pokkesviricetes, and asks how the genes with introns were treated. Reviewer #2 wants you to rewrite some sentences for clarification, says that you should provide information about the source of the genomes and about the viruses contributing to each branch of the tree, asks whether there are 8 or 9 or 7 marker gene sets, thinks that you should differentiate branches containing viruses derived from culturable isolates vs. viruses assembled from metagenomic data, and wants you to add a supplementary table that provides the complete set of RED values along with the identification of the nodes/clades associated with each value. Reviewer #3 thinks that you should add gene names to the supplemental figures S1-25 and wants to see the trees associated with the combined marker genes in Fig.1. This reviewer also thinks that it would be helpful for communication if names were given at the lower levels too when possible.

DATA POLICY:

Regardless of the method selected, please ensure that you provide the individual numerical values that underlie the summary data displayed in the following figure panels as they are essential for readers to assess your analysis and to reproduce it: Figures 1AB, 3A, S26.

**Please also ensure that figure legends in your manuscript include information on where the underlying data can be found, and ensure your supplemental data file/s has a legend.**

We expect to receive your revised manuscript within 1 month.

**IMPORTANT - SUBMITTING YOUR REVISION**

*Resubmission Checklist*

*Published Peer Review*

*PLOS Data Policy*

Sincerely,

Paula

---

Paula Jauregui, PhD

Associate Editor

PLOS Biology

REVIEWS:

Reviewer #1: Hiroyuki Ogata. Evolution and ecology of Giant viruses.

Reviewer #2: Virus taxonomy.

Reviewer #3: Giant viruses in soil.

Reviewer #1: Viruses of the phylum Nucleocytoviricota are known for their large genomes, particles, and intriguing biological traits and potential ecological roles. The ICTV taxonomy of this viral group had been incorrect in many aspects for long time until when the recent viral megataxonomy was established in 2020, by a group of leading virologists/taxonomists including one of the authors of this manuscript. However, recent metagenomics has started to produce hundreds to thousands of additional environmental genomes of the Nucleocytoviricota. Consequently, the need for updated phylogenomic studies have re-emerged in order to establish common languages on the taxonomies within the phylum including those uncultivated. As authors correctly pointed out in their manuscript, this updated taxonomic framework is important for ecological and evolutionary analyses of this group of viruses in connection with their relationships with the evolution of cellular organisms. This is very timely and useful work.

The authors compiled 1380 Nucleocytoviricota viral genomes (both isolated and uncultured ones), identified 8863 protein families (named GVOGs), of which they identified 25 GVOGs that are represented widely in the genomes. I have appreciated the next steps that authors took to identify 7 marker genes that are considered as the best choice for phylogenomic analyses, and to objectively delineate clades (class, order, family, and genus level). The choice of marker genes for the phylogenomics is based on the Tree Certainty (TC) scores, while the taxonomic delineation has been performed using a procedure based on the Relative Evolutionary Distance (RAD). To my opinion, this is the best plausible strategy to delineate the taxonomic clades of this viral group. With these methods, the authors partitioned 2 classes, 6 orders, 32 families (this high number is appropriate), and 344 genera within the phylum. The following descriptions of the compositions of each order are easy to read and well-balanced with previous taxonomic proposals. I have nevertheless a few comments that I hope would help to further clarify some points in the current version of the manuscript.

L.33: It would be useful to provide the number of currently recognized viral genera in the phylum Nucleocytoviricota.

LL.48-54: The citation of the references 9,15,20 appear twice in this part, which makes the logical flow awkward. Probably, the first citation of "9,15,20" would not be needed.

L.75: The definition of the word "megataxonomy" may be unclear to some readers (although it is a title of well-cited paper). A few words explaining this jargon would be useful.

L.103: "this is likely the case because … multiple copies of MCP, … orthologs from paralogs (Fig. 1a)." To my understanding, for tree reconstructions, the authors chose one sequence for multicopy genes from each genome based on the score against respective HMMs. If it is the case, one MCP at most was chosen from each genome. If this is how data was processed, then, I do not agree to attribute the low TC value for the MCP tree to the existence of paralogs. Because the choice of paralogs or orthologs does not affect the reliability of their tree (if paralogs are mixed, then the tree would not be a species tree but still a gene tree). I guess it is more due to the variable nature of MCPs. Related to this, LL.365-366 for the chose of sequence dataset for individual families is somehow unclear. Does this include only the best hits, or multiple hits if they pass the threshold? A few additional words may help readers to completely understand the procedure.

L.119: The fact that the addition of RNAPS led to the decreased TC may be due to the existence of paralog. In this case, because the goal is to gain the consistency among species trees, the choice of ortholog is needed. RNAPS paralogs was described in https://journals.asm.org/doi/10.1128/JVI.00230-17.

L.140: The computation of RED requires the root of the whole tree to my understanding. The authors mentioned that Pox-Asfar group was used to root the tree. However, this would not give the root of the whole phylogenetic tree. How was the root defined? Mid-point? This should be described in the Method section.

L.154: "including 213 genera with a single representative". Just above this statement, the authors state that clades with two or one members were left as singletons, and for the family count (n=32), the authors excluded the singletons. Consistent counting scheme would be easy to follow.

L.165: The citation of ref-22 should be ref-47.

LL.173-192: The proposed classification scheme does not refer to the Klosneuvirinae, which are also used in several published papers. I suggest the authors to describe their standpoint about this taxonomic group, so that the readers can judge if they wish to continue using this taxonomic group name.

L.199-202: Citations of the papers that described the discoveries of individual genes would be needed. For actin, myosin, and kinesin, the citations seem to be https://doi.org/10.1101/2020.06.16.150565, https://doi.org/10.3389/fmicb.2021.683294, doi: 10.1128/mSystems.00293-21.

Fig.4: The enrichment of Translation genes in the PM_09 is striking but this is not mentioned in the text. A few words on this may be useful for readers.

L.240-241: In the ref-45, the PolB of medusavirus was proposed to be transferred to eukaryotes, whereas the authors assume a transfer from eukaryotes to medusavirus. These should be distinguished.

L.263-272: The authors cite two genomes in the Pimascovirales and three genomes in the Pokkesviricetes. First of all, the branches that correspond to these five genomes should be clearly indicated in the Fig.2. In addition, if the two genomes in the Pimascovirales correspond to the two branches below the Mininucleoviridae, then they look like a family level clade (within the Pimascovirales) instead of an order level clade. If these two genomes form a new order, Mininucleoviridae should be also placed at the order level too. Additional explanation is needed.

L.333: The author used Prodigal for gene call. How were genes with introns treated? This should be clear as some of the genomes contain many introns.

Reviewer #2: The work described in this manuscript to develop a framework for studying the diversity and evolution of large DNA viruses, provides an extremely useful and valuable set of resources (the giant virus orthologous group database along with associated alignments and trees) and data that help to further classify these viruses with members that have either been physically isolated or identified via metagenomic sequencing of environmental samples. As such, it represents an important contribution to the literature. The manuscript will benefit from a careful revision to correct a number of problems, including those detailed below. Major problems include failure to include Table 1, and a misnumbering of tables in the text. In addition, it will be extremely useful to have more of the original data generated in support of this work, available to its readers.

Lines 54-55: The initial wording of the sentence beginning "The uncertainty of the phylogenetic structure and taxonomy Nucleocytoviricota…" needs rewriting.

Line 83: Table S1 is not a list of genomes. This reference should most likely be to Table S2. Table S2 should provide traditional GenBank accession numbers for all genome sequences when available. When GenBank accessions are not available, the source of the indicated genome sequence (and the provided genome_id) should be identified. I have no idea what some of the provided genome_ids refer to. (I would recommend not concatenating different IDs together.)

Line 89: There is no Table 1 provided with the manuscript.

Lines 311,312: Four sequences are mentioned that are not in an NCBI database. If these sequences are not readily available, they should not be included in this analysis.

Line 90: Table S1 does not provide, as indicated in the text, a "descriptions of all GVOGs".

Line 377: There is no Table 1

Line 387: Are there eight or nine marker gene sets?

Line 401: Now we are at (the final) 7 marker sets. The number of marker gene sets varies according to the analysis providing in figure 3. But in this section of the Methods, it is not clear why these numbers vary from 9 to 8 to 7.

In the supplementary phylogenetic trees for the set of 25 gene sets, it is very difficult to determine from the provided figures the characterizations of the sets provided in Table S1. For example, for GVOGs 115 and 152, the table indicates that these GVOGS are not represented in viruses belonging to the order Asfuvirales. In the figures for these two GVOG trees, there is also no indication of the presence of poxviruses (order Chitovirales) in the tree. GVOG 152 also appears to not have a representative from the pandoraviruses.

In all text and figures, the proposed order pandoravirales should not be capitalized or italicized since it is only proposed and is not an official taxon approved by the ICTV.

Figure 2 and corresponding text: All proposed taxa should, as indicated above, not be capitalized or italicized since they are not official taxa approved by the ICTV.

Figure 2: It would be useful to be able to identify the viruses contributing to each branch of the tree, especially for unlabeled, single genome branches. It is not clear if this information is available or where it might be.

Figure 2: It would be useful to differentiate branches containing viruses derived from culturable isolates vs. viruses assembled from metagenomic data.

Figure 3a: A supplementary table should be available that provides the complete set of RED values along with, importantly, the identification of the nodes/clades associated with each value. Along with Table S2, this will provide the data that supports the creation of the proposed new species and families.

Figure 4. Panels A and B should be labeled on the figure.

Reviewer #3: Recently several metadata studies (including one by these authors) have identified thousands of new genomes in the Nucleocytoviricota from environmental samples. This current work provides a phylogenetic and more importantly a taxonomic context to interpret and communicate this data. The research uses appropriate state of the art methods. An important aspect is the used of RED values to normalize taxonomic levels to ranges of genetic distances. The manuscript is very well written and will be an important contribution to the field. I only have minor concerns.

In browsing the 25 GVOG trees is the supplemental data S1-25, it was clear that different trees gave very different results. It would be nice to add gene names to the supplemental figures S1-25.

It would be good to see the trees associated with the combined marker genes in Fig.1 (maybe I missed these?). How did they vary? This would give a sense of how robust the taxonomy is to any individual gene.

Reference in the manuscript the itol tree https://itol.embl.de/tree/1281731864487941620067021 on the github site.

The authors only suggested names at the phylum, class and order level. All families and genera were given numbers. It would be helpful for communication if names were given at these lower levels too when possible or is the expectation that these will need to be dealt with separately as many genera will need to be renamed as species.

---

## [Editor Report · Decision Letter 2]

15 Sep 2021

Dear Dr. Aylward,

Thank you for submitting your revised Methods and Resources entitled "A Phylogenomic Framework for Charting the Diversity and Evolution of Giant Viruses" for publication in PLOS Biology. I have now discussed your revision with the Academic Editor. 

We will probably accept this manuscript for publication, provided you satisfactorily address the following data and other policy-related requests.

DATA POLICY:

Regardless of the method selected, please ensure that you provide the individual numerical values that underlie the summary data displayed in the following figure panels as they are essential for readers to assess your analysis and to reproduce it: Figures 1AB, 3A, S26.

**Please also ensure that figure legends in your manuscript include information on where the underlying data can be found, and ensure your supplemental data file/s has a legend.**

We expect to receive your revised manuscript within two weeks.

*Published Peer Review History*

*Early Version*

Sincerely,

Paula 

---

Associate Editor,

pjaureguionieva@plos.org,

PLOS Biology

---

## [Editor Report · Decision Letter 3]

29 Sep 2021

Dear Dr Aylward,

I'm handling your paper temporarily while my colleague Dr Paula Jauregui is out of the office. On behalf of my colleagues and the Academic Editor, Curtis Suttle, I'm pleased to say that we can in principle offer to publish your Methods and Resources "A Phylogenomic Framework for Charting the Diversity and Evolution of Giant Viruses" in PLOS Biology, provided you address any remaining formatting and reporting issues. These will be detailed in an email that will follow this letter and that you will usually receive within 2-3 business days, during which time no action is required from you. Please note that we will not be able to formally accept your manuscript and schedule it for publication until you have made the required changes.

PRESS: We frequently collaborate with press offices. If your institution or institutions have a press office, please notify them about your upcoming paper at this point, to enable them to help maximise its impact. If the press office is planning to promote your findings, we would be grateful if they could coordinate with biologypress@plos.org. If you have not yet opted out of the early version process, we ask that you notify us immediately of any press plans so that we may do so on your behalf.

Sincerely, 

Roli Roberts

Roland G Roberts PhD

Senior Editor

PLOS Biology

rroberts@plos.org

on behalf of

Paula Jauregui, PhD 

Associate Editor 

PLOS Biology
